# The Application of Hybridization Chain Reaction in the Detection of Foodborne Pathogens

**DOI:** 10.3390/foods12224067

**Published:** 2023-11-09

**Authors:** Jinbin Zhao, Yulan Guo, Xueer Ma, Shitong Liu, Chunmeng Sun, Ming Cai, Yuyang Chi, Kun Xu

**Affiliations:** 1School of Medicine, Hunan Normal University, Changsha 410013, China; jinbinzhao125@163.com; 2Department of Nutrition and Food Hygiene, School of Public Health, Jilin University, Changchun 130021, China; 3The Research Center of Reproduction and Translational Medicine of Hunan Province, Changsha 410013, China

**Keywords:** hybridization chain reaction, foodborne pathogens, food safety, biosensor

## Abstract

Today, with the globalization of the food trade progressing, food safety continues to warrant widespread attention. Foodborne diseases caused by contaminated food, including foodborne pathogens, seriously threaten public health and the economy. This has led to the development of more sensitive and accurate methods for detecting pathogenic bacteria. Many signal amplification techniques have been used to improve the sensitivity of foodborne pathogen detection. Among them, hybridization chain reaction (HCR), an isothermal nucleic acid hybridization signal amplification technique, has received increasing attention due to its enzyme-free and isothermal characteristics, and pathogenic bacteria detection methods using HCR for signal amplification have experienced rapid development in the last five years. In this review, we first describe the development of detection technologies for food contaminants represented by pathogens and introduce the fundamental principles, classifications, and characteristics of HCR. Furthermore, we highlight the application of various biosensors based on HCR nucleic acid amplification technology in detecting foodborne pathogens. Lastly, we summarize and offer insights into the prospects of HCR technology and its application in pathogen detection.

## 1. Introduction

With the continuous development of international food trade and the globalization of the food market, food safety is still worthy of extensive attention. As we all know, foodborne pathogens (bacteria, fungi, and viruses) widely exist in human daily life. They quickly cause foodborne diseases by contaminating food in food production, processing, storage, and transportation, which poses a significant threat to public health, personal health, and the national economy [1,2]. According to the World Health Organization (WHO), foodborne diseases cause 600 million cases and 420,000 deaths worldwide each year, of which 30% of foodborne deaths occur, especially in 5-year-old children, and the loss of 33 million disability-adjusted life years (DALY) [3,4]. The most common foodborne pathogens include *Salmonella. Enteritidis* (*S. enteritidis*), *Salmonella Typhimurium* (*S. Typhimurium*), *Campylobacter*, *Staphylococcus aureus* (*S. aureus*), *Listeria monocytogenes* (*L. monocytogenes*), and *Escherichia coli O157:H7* (*E. coli O157:H7*) [5,6,7]. It is worth noting that some bacteria and fungi produce toxins during their growth and reproduction, and even many of these pathogens and toxins are thermostable, and therefore cannot be easily destroyed by typical food preparation methods (cooking, frying, freezing, etc.) [8]. Early detection of foodborne pathogens is necessary to prevent outbreaks of foodborne diseases, improve overall cost-effectiveness, and ensure better food safety management [9].

For the detection of pathogenic bacteria, the traditional culture method needs to go through the procedures of pathogenic bacteria cultivation, isolation, recovery, and identification, which often takes 3–7 days and has the disadvantages of requiring professional staff, being time-consuming, and having a complicated operation process, as well as being a false negative and lacking sensitivity, which is challenging to adapt to the development of rapid, accurate, and sensitive detection technology. With the development of immunology and molecular biology, enzyme-linked immunosorbent assays (ELISA) and polymerase chain reaction (PCR) have become the gold standard for detecting foodborne pathogens. However, both methods have their drawbacks. ELISA, which is based on the immune response, is limited by the complexity of antibody preparation and lot-to-lot variations. Based on nucleic acid amplification, PCR has high sensitivity but requires specialized personnel and expensive equipment and is prone to cross-contamination and false positives [10,11,12,13]. The development of timely, sensitive, accurate, and cost-effective technologies for detecting foodborne pathogens has therefore been a challenging task in food safety. In order to achieve the sensitivity required to detect pathogens at deficient concentrations in complex food matrices, signal amplification techniques such as enzyme-linked cascade amplification and nucleic acid amplification [14] are increasingly developing and applied for pathogen detection to meet the growing demand for food safety.

Some bacteria (such as emetic Bacillus cereus) are non-antigenic and lack antibodies. In addition, proteins themselves cannot be amplified, such as antibodies or antigens. Therefore, nucleic acid-based signal amplification has more significant potential advantages and generally falls into two categories: thermal cycle amplification techniques, such as PCR and ligase chain reaction (LCR) [15]; and isothermal amplification techniques, including rolling circle amplification (RCA), loop-mediated isothermal amplification (LAMP), strand displacement amplification (SDA), hybridization chain reaction (HCR), recombinase polymerase amplification (RPA), etc. Isothermal amplification is an emerging nucleic acid amplification method attractive to many food production industries due to its constant temperature, cost-effectiveness, and the absence of expensive thermal cycling equipment in thermal cycle amplification techniques. In particular, the simple, enzyme-free, and easy-to-modify nature of HCR has made it an increasingly popular signal amplification tool in biosensing, bioimaging, and biomedical applications. In recent years, the application of HCR to bacterial detection has emerged like mushrooms after rain, further promoting the development of foodborne pathogen detection technology towards simplicity, sensitivity, and on-site detection.

## 2. Overview of HCR Technology

HCR, first proposed by Dirks and Pierce in 2004, is an enzyme-free, entropy-driven, instrument-independent isothermal amplification technique with simultaneous recognition and signal amplification capabilities, producing ladder-shaped products with fragments of different sizes, the molecular weight of which is usually inversely proportional to the size of the initiator [16]. Essentially, it is a toehold-mediated strand displacement reaction (TMSD) initiated by a trigger or target molecule [17]. 

### 2.1. Principles and Types of HCR Technology

There are two main types of HCR: linear HCR and non-linear HCR [18]. Typical linear HCR is today’s most widely used type because of its high versatility and good amplification capability. It consists of three strands: an initiator strand and a pair of complementary paired fuel strands (H1, H2) with sticky ends. The potential energy in the ring is locked by the “stem-loop” structure in the hairpin probes, which can exist in a relatively stable state without the initiator. When the initiator is present, the sticky end of the H1 chain of the fuel strand pairs with the initiator, releasing energy, then pairs with the sticky end of the H2 chain and exposes the same sequence as the initiator, causing the two fuel strands to cross-hybridize and self-assemble into a long gap-containing double-stranded DNA (dsDNA) polymer nanowire with repeating [H1-H2-] units, indirectly amplifying the target chain (Figure 1). However, it suffers from easy leakage and low sensitivity due to low target abundance and environmental complexity.

Piece et al. 2007 proposed an improved type of migrating HCR that significantly reduces the possibility of non-specific initiation and relaxes sequence design requirements. However, there is still the problem of low sensitivity due to the limited amplification of linear HCRs. For this reason, continuous efforts have been made to develop non-linear HCR types with their characteristics. Some are named after the unique hairpin design and generation of high-molecular products, such as dumbbell HCR (DB-HCR), branched HCR (B-HCR), and dendritic HCR (D-HCR); some are based on the characteristics of the application, such as multiplex HCR (multi-HCR), with a large surface area of nanoparticles loaded with a large number of starting chains, and in situ-HCR for bioimaging. In addition, it is worth noting that the efficiency of HCR hybridization in a solution environment is influenced by the collision rate of molecular monomers at a local scale. Therefore, in addition to the focus on improving HCR design to increase HCR sensitivity, several methods to increase collision frequency by appropriately increasing probe concentration and using unique DNA structures (e.g., DNA origami, DNA tetrahedra, etc.) and multi-technology combinatorial strategies in the form of “HCR + X” are also being developed, driving the broader development of HCR.

### 2.2. HCR Technical Characteristics

HCR technology has its unique attractions and features. The primary feature is that it generates long dsDNA polymers with repeating units for signal amplification based on the extension of the target sequence rather than making copies of the target, which avoids the problems of false positive results and cross-contamination that often occur with PCR, LAMP, and other techniques using universal primers [19]. Secondly, the non-targeted amplification approach allows for the prior preparation of HCR products, and thus can reduce the assay time. Finally, the signal molecules obtained by most amplification methods are challenging to immobilize. In contrast, the signal molecules generated by HCR can be immobilized in some solid phases (e.g., magnetic beads, cell surfaces) by anchoring the initiation chain, thus enabling the collection and localization of signals.

## 3. HCR in Foodborne Pathogen Detection

The ideal pathogen detection method should be rapid, sensitive, specific, economical, equipment-free, and intuitive. Therefore, developing new detection methods that meet the comprehensive performance requirements of quantification, sensitivity, specificity, and economy is urgently needed. Among them, HCR has attracted much attention due to its advantages, such as being enzyme-free, having isothermal conditions, its self-assembly, and excellent amplification efficiency. In addition, new technology for pathogen detection often needs to have three parts: identification, transduction or amplification, and output signal. Among them, HCR is based on the specific recognition and triggering characteristics of target nucleic acid and has both easy modification of signal molecule transduction (such as electrochemical reagents [20], nanoparticles [21], fluorescent dyes [22], biotin, etc.) and good amplification ability, so that it can easily combined with various signal output platforms (such as colorimetry, fluorescence, electrochemistry, Raman scattering, chemiluminescence, etc.), which has excellent potential for further expansion in the field of foodborne pathogen detection [23,24,25,26].

### 3.1. Colorimetry

Colorimetry is a typical biochemical assay that detects analytes by comparing or measuring the color change caused by a specific reaction between the analyte and the sensing material, either by the naked eye or by a simple optical instrument, which allows for intuitive qualitative or quantitative detection [27,28,29]. The colorimetric method is widely used because of its simplicity, rapidity, cost-effectiveness, and visualization of results [30]. Colorimetric results usually include changes in a single-color shade and color type. The first type is represented by enzymes (e.g., natural enzymes, nano enzymes, DNA enzymes), which cause changes in single color intensity due to the difference in the catalytic ability of the enzyme on the chromogenic substrate; the second type is represented by nanoparticles (gold and silver nanoparticles, nanoclusters), which cause a change in color through changes in extinction coefficient caused by aggregation, growth, etching, etc. However, the sensitivity of colorimetric sensing systems is sometimes unsatisfactory [31]. Considering improvements by introducing technical solutions such as HCR signal amplification to achieve sensitive detection is more than welcome.

In 2016, the Lai group first reported a sandwich ELISA for detecting *Escherichia coli O157:H7 (E. coli O157:H7)* in milk using HCR loaded with horseradish peroxidase (HRP) [32]. A gold nanoparticle probe (mAb-AuNP-DNA1) modified with specific antibodies and trigger chains was used as a signal transduction and amplification carrier. In the presence of *E. coli O157:H7*, a “pAb/*E. coli O157:H7*/mAb-AuNP-DNA1” sandwich complex was formed on the ELISA substrate, followed by the addition of biotin-modified hairpin probes (H1-biotin, H2-biotin). Triggered by DNA1, HCR generated long-nicked dsDNA rich in biotin sites. After the addition of horseradish peroxidase (HRP-SA) anchored with streptavidin and chromogenic substrate (TMB + H_2_O_2_), this sandwich ELISA method had a detection limit of 1.08 × 10^2^ CFU/mL in the concentration range of 5 × 10^2^–1 × 10^7^ CFU/mL of *E. coli O157:H7*, which was 185 times lower than that of the traditional ELISA (Figure 2A).

Furthermore, not only can nucleases (DNAzyme) solve the unstable and expensive problems of natural enzymes and the editable properties of nucleic acids with the compatible rivalry of HCR’s editable nature, but HCR-DNAzyme can be developed into a universal paradigm. Tian et al. proposed an ultrasensitive colorimetric dual-target detection platform based on HCR-G-quadruplex/hemin DNAzyme (HCR-GQH DNAzyme) [33]. First, this method allows for multiplex super PCR (MS-PCR) simultaneous dual-target amplification—the reverse primer labeled biotin for binding with magnetic probes to eliminate primer dimer interference. In contrast, the forward primer is blocked by inserting an ethylene glycol bridge inhibitor to limit the extension of the polymerase, resulting in a large number of dsDNA products with single-strand DNA (ssDNA) toes protruding (ssDNA-dsDNA). Then, two pairs of hairpin probes containing 5 nt oligonucleotide tails (H1, H2, and H3, H4) perform their respective HCR reactions, producing many HCR products containing 5 nt oligonucleotide tails. The interference hairpin is removed by magnetic separation, and Terminal deoxynucleotidyl Transferase (TdT) catalyzed 5-nt ssDNA tail extension to amplify, then form GQH DNAzyme, which catalyzes the oxidation of ABTS to produce a green color. This visual detection method creatively combines MS-PCR and AT-HCR-DNAzyme to simultaneously detect Salmonella and Staphylococcus aureus in fat-free milk samples with a 10 CFU/mL detection limit (Figure 2B). 

In addition, AuNP is also popular for its property of changing the color of the solution due to aggregation and dispersion. However, the sensitivity of sensors based only on unlabeled aptamers and AuNP has been reported to reach only 10^5^ CFU/mL [35]. Therefore, it is urgent to improve the detection sensitivity of such colorimetric methods by increasing the reaction efficiency, cycling amplification [36,37], or introducing signal amplification technologies such as HCR. 

Sun et al. innovatively combined two mechanisms to develop a highly sensitive dual-signal amplification colorimetric strategy [34] based on DNA@AuNPs (Probe 1 and Probe 2/3) aggregation, in which Probe 1 is formed by the interaction between AuNPs and sticky-ended short ssDNA, and Probe 2/3 utilizes thiolated DNA to stabilize AuNPs via “Au-S” bonds (Figure 2C). When the target is present, the polymer releases DNA 1, which activates the HCR, and thus induces the aggregation of Probe 1. Meanwhile, DNA 3, as a linker, separates from Probe 1 and further assembles Probe 2/3. The more sensitive color response of the system is due to the synergistic aggregation of two different DNA@AuNPs probes triggered by one activation. Under optimal conditions, the ultrasensitive AuNPs-based strategy achieved sensitive detection of 10 CFU/mL and was successfully applied to detect *E. coli* from tap water and milk samples. 

However, single-color colorimetric methods continue to face the problem of being difficult to distinguish and having low color resolution. With the development of nanoscience, various AuNP-derived structures have emerged (such as gold rods, gold nanobipyramids, gold nanostars, silver nanoprisms, etc.) that break the shackles of a single color. Due to their sharp morphology, small changes at hot spots easily lead to sensitive changes in extinction coefficients, resulting in rainbow-like color changes, promoting the broader development of colorimetric methods.

Tang et al. reported a high-color-resolution and ultrasensitive detection biosensor [38] for the detection of methicillin-resistant Staphylococcus aureus (*mecA* gene) in milk based on the multi-HCR of AuNPs and the alkaline phosphatase (ALP)-mediated in situ growth of gold nanobipyramid particles (AuNBPs) (Figure 3A). In the presence of target DNA, it acts as a linking bridge to hybridize magnetic capture probes (CP) and AuNP signal probes (SP), carrying plenty of ALP. With the assistance of NADPH, a series of rainbow-colored results of the in situ growth of AuNBPs with different concentrations of target DNA are observed. This method can detect as low as 2.71 pM target DNA through multi-step amplification of AuNP vectors, multi-HCR amplification, and enzymatic reactions. Importantly, the multiple color output results are easy to determine visually.

At the same time, the powerful editable capability of HCR not only significantly enhances specificity but also enables cascade amplification of the enzyme. Yang et al. by using a DNA logic gate strategy and combining Dig-Probe and FITC-probe strategies (Figure 3B). Only when three specific regions of the *L. monocytogenes* specific genomic sequences are present simultaneously is the HCR reaction triggered to form a long dsDNA that is enriched with glucose oxidase (GOx) and is HRP-rich. GOx converts glucose to gluconate and produces hydrogen peroxide, allowing it to further mediate the oxidation of TMB by HRP, producing a highly specific visual color signal output for enzyme cascade amplification with a detection limit of 1.12 nM [39], and successfully applied to a variety of food matrices such as vegetables, meat, and milk extracts.

### 3.2. Fluorescence

Fluorescence (FL)-based methods have a higher sensitivity compared to colorimetric methods, with fluorescence spectra as the resultant response. The current fluorescence strategies using HCR for pathogen detection are broadly divided into two types [40]: (1) labeling of fluorescent groups or insertion of fluorescent dyes; (2) fluorescence energy transfer, including fluorescence resonance energy transfer (FRET) and contact quenching (CQ) [18]. 

The first type is more cost-effective than the others and is commonly used, but the sensitivity is somewhat reduced by the use of a single fluorescent group or dye with its high background fluorescence interference. Therefore, it is often necessary to separate or adsorb the background fluorescence with the aid of various nanomaterials (e.g., magnetic beads, graphene oxide, MnO_2_, MXene,) to obtain an accurate and sensitive fluorescence signal.

Yu et al. proposed a fluorescent sensor for detecting *Salmonella* (*S. Enteritidis*, *S. Typhimurium*, and *S. Choleraesuis*) *invA* gene in lettuce based on HCR using MBs as a separator [41]. Long ssDNA prepared by asymmetric polymerase chain reaction (aPCR) was subsequently captured by capture probe-modified MBs, the exposed target ssDNA triggered HCR amplification of FAM-labelled reporter probes (H1-FAM, H2-FAM), and finally, magnetic separation and boiling released the reacted hairpin reporter probes, achieving a limit of detection of 7.4 × 10^1^ CFU/mL in buffer and 6.9 × 10^2^ CFU/g in spiked lettuce (Figure 4A). There is also some research using a combined strategy of magnetic beads and HCR, resulting in low-cost, reusable recycling [42] and high-throughput detection [43].

Apart from MBs, the two-dimensional nanomaterial—graphene oxide (GO) [45] is commonly preferred for its excellent fluorescence quenching properties for most fluorescent dyes, fluorescent groups, quantum dots, and metal fluorescent nanoclusters. In addition, it has been cleverly used as an adjunct to fluorescence strategies in HCR technology due to its different affinity for ssDNA and dsDNA and as an “HCR-GO” platform to reduce background values or restore fluorescence signals. Yu et al. proposed a fluorescence “turn-on” strategy based on the “HCR-GO” platform [44]. Initially, the hairpin single strands were quenched by GO, and fluorescence was turned off. When *Salmonella* ssDNA was present, HCR was triggered by the alternating hybridization of H1-FAM and H2-FAM to form a long dsDNA nanowire, which detached from the GO surface and restored fluorescence. The detection of *Salmonella* ssDNA was completed within 3.5 h, with a detection limit of 4.2 × 10^1^ CFU/mL in buffer and 4.2 × 10^2^ CFU/mL in milk (Figure 4B).

With the aid of nanotechnology, various HCR-based assisted co-amplification strategies (e.g., multi-technology co-amplification, enzyme-assisted HCR, fluorescent synergy effect, etc.) have been proposed. Liang et al. coupled variable temperature amplification with isothermal amplification techniques to establish a novel PCR-HCR dual-signal amplification fluorescence method [46], effectively increasing the sensitivity of the assay to 100 times that of PCR signal amplification alone, with a detection limit of 7.2 × 10^1^ CFU/mL in buffer and 7.2 × 10^2^ CFU/mL in milk (Figure 5A). Zhang et al. coupled two isothermal amplification techniques to synergistically amplify the detection signal and developed a DNA fluorescence sensor combining HCR and three-way cleavage-assisted signal amplification (3WJ-NEASA) for dual-signal amplification for the first time [47]. The target DNA of *S. aureus* initiates self-assembly between H1 and H2, exposing the gap to capture molecular beacon to form the 3WJ structure, which is continuously cycled by Nt.BbvcI cleavage enzymes to cleave and capture the molecular beacon, resulting in the accumulation of recovered fluorescence. The benign cycle of the HCR circuit and the 3WJ-NEASA circuit amplifies synergistically, allowing the biosensor to detect *S. aureus* DNA down to 6.7 pM in one step within 30 min. Finally, the HCR-mediated 3WJ-NEASA method detects *S. aureus* with a low LOD of 1.2 × 10^1^ CFU/mL and 1.3 × 10^2^ CFU/mL in milk (Figure 5B).

In addition, the HCR technique can easily combine dsDNA dyes (e.g., SYBR Green I) and FAM fluorophores to produce a fluorescence synergy effect to enhance the fluorescence signal due to the generation of double-stranded products and ease of modification. Tang et al. combined the “FAM-SYBR Green I” fluorescent synergy effect with the “HCR-GO” platform and achieved sensitive detection of *S. aureus* 16S rRNA with a LOD of 50 pM (Figure 5C). It was also successfully applied to detect milk samples with a LOD of 4 × 10^2^ CFU/mL [48]. The Lu group proposed two ultrasensitive fluorescent detection strategies for methicillin-resistant *S. aureus* mRNA based on the synergistic effect of “FAM-SYBR Green I” on the “HCR-GO” platform and further incorporating other cyclic amplification strategies [49,50].

FRET can effectively reduce its background fluorescence by bringing the donor-acceptor fluorophores closer together without the aid of additional nanomaterials, which can be easily achieved by HCR self-assembly. Ren et al. proposed a simple FRET-based fluorescence sensor for the detection of *Vibrio parahaemolyticus* (*V. parahaemolyticus*) ssDNA using a four-way migration HCR [51]. When ssDNA is present, the auxiliary strand (R) first binds to the target strand to form short dsDNA, which triggers the four-way migration of HCR, and FRET occurs. The method is capable of detecting 0.067 nM ssDNA (Figure 6A). In addition to the use of HCR self-assembly to close the distance between fluorescent groups, FRET-sensitive *V. parahaemolyticus* detection strategies have also been investigated based on the formation of dsDNA by triple oligonucleotide (TFO) insertion [52]. 

The ratio fluorescence output is another feature that makes FRET more popular, thus greatly reducing the interference of uncontrollable factors such as light scattering and photobleaching and obtaining more accurate results. Cai et al. [53] used the “IMB/target/AuNP” sandwich structure to achieve bacterial cell signal converted and amplified into a biological barcode DNA molecule, which triggers the trans-cleavage activity of CRISPR-Cas12a, cleaving the initiator chain of TDN-hHCR, resulting in the failure of the HCR product formation and no FRET occurring between hairpin fluorophores. In contrast, in the absence of *Salmonella*, CRISPR-Cas12a cannot be activated, and the initiator is intact, triggering the TDN-hHCR process and promoting FRET. This method has been applied to actual samples, achieving *Salmonella* sensitive detection of 17 CFU/mL in milk and 25 CFU/mL in egg white (Figure 6B).

CQ is another fundamental mechanism of fluorescence energy transfer. When the fluorescent group and the quencher are in proximity in this system, the energy of the fluorescent group can be transferred to the quencher through proton-coupled electron transfer, leading to fluorescence contact quenching [55]. As the distance between them increases, the fluorescence on the fluorescent group can be restored, resulting in a fluorescent signal. The Lai group coupled CQ and ratio FRET through HCR technology, proposing a sandwich ELISA method for the detection of *E. coli O157:H7* in milk. They designed a low-background HCR hairpin probe (CQ-FRET probe) to investigate a more sensitive and accurate FRET output strategy [54]. The detection limit of this CQ-FRET method was 3.5 × 10^1^ CFU/mL, significantly lower than that of the CQ hairpin-based immune HCR (3.28 × 10^3^ CFU/mL) and the FRET hairpin-based immune HCR (6.49 × 10^4^ CFU/mL), which were 93-fold and 1854-fold higher, respectively. But there is no doubt that it also increases costs and expenses (Figure 6C).

Finally, it is noteworthy that most of the current HCR-based fluorescence sensing strategies are downconverted fluorescence strategies using high-energy ultraviolet as the excitation light source, which have some obvious drawbacks such as photobleaching, background auto-fluorescence, and cell-damaging properties, limiting their real-time applications. Near-infrared (NIR)-excited upconversion fluorescence strategies represented by lanthanide-doped upconversion nanoparticles (UCNPs) are receiving much attention in the fields of cell imaging and tumor therapy [56]. Recently, the upconversion strategy has also been applied to HCR-based fluorescence detection of foodborne pathogens [57], where the HCR technique amplifies the strongly stabilized fluorescence signal, in addition to the NIR excitation mode, which also has the potential to penetrate deeply into food monitoring.

### 3.3. Electrochemistry

Electrochemical biosensors typically employ three main kinds of signals as their readout, namely current, voltage, and impedance signal, to reflect the concentrations of analytes [58,59,60,61]. Due to its excellent signal amplification and easy labeling, HCR can load with catalytic elements (e.g., natural enzymes [62], nanoenzymes [63], etc.) or conductive substances (e.g., redox molecules [64], conductive nanomaterials [65], etc.) through modification, electrostatic adsorption, or embedding to influence the electrical signal transduction capacity of the electrode [66,67]. Electrochemical approaches with HCR as the signal amplifier have already attracted more and more attention in bioassays due to their excellent sensitivity as low as the attomolar level [68].

A sensitive electrochemical strategy for the detection of *S. aureus* in milk and pear juice based on DB-HCR was proposed by Wu et al (Figure 7A). The use of DB-HCR to produce dense products is a perfect solution to the problem of loss of sensitivity in electrochemistry that is often caused by the production of long linear products that deviate from the electrode in conventional HCR. The presence of a target leads to the release of the T-strand from the “Apt-T” complex and its capture by the electrode surface capture probe, triggering the DB-HCR on the electrode surface to generate a dense negatively charged DNA product that attracts a bulk loading of modified positively charged poly methylene blue nanoparticles (pMBNPs), which achieves ultrasensitive detection down to 1 CFU/mL and demonstrates the excellent amplification potential of DB-HCR in electrochemical sensors [69].

Li et al. proposed an ultrasensitive electrochemical biosensor based on 3D DNA Walker, RCA, and HCR multiplex amplification [70]. By generating dendritic DNA products by HCR, the spatial site resistance problem, often present in electrochemical receptors, is avoided. This assay demonstrated high sensitivity to the target DNA, with a 7 CFU/mL LOD (Figure 7B).

In addition, Feng et al. cleverly exploited the characteristics of HCR-grown linear DNA nanowires to propose a simple and novel electrochemical scheme of simple synthetic silver wires between electrodes to detect *S. aureus* 16S rRNA in milk [71]. In the presence of 16S rRNA, the modified sub-stable hairpin probe H1 on the electrode surface opened, activating HCR. The stem-loop structures of H1 and the hairpin probe H2 modified with AuNPs (H2-AuNPs) alternately opened to finally form a long dsDNA-RNA (HCR product)-AuNPs product. Under the influence of N_2_ cycles, the modified HCR product of the AuNPs was tiled in the electrode gap to form a framework, and silver was deposited on the AuNP of the extended ladder product between the electrodes to form silver wires, leading to a drastic change in electrical parameters. This method achieved a detection limit of 50 CFU/mL within 50–10^7^ CFU/mL of *S. aureus*, with a total time of 100 min (Figure 7C).

In recent years, the clustered regularly interspaced short palindromic repeats (CRISPR) and CRISPR-associated proteins (Cas) systems have been widely reported in the field of biosensing due to their high-turnover non-specific endonuclease activity [73,74]. However, the sensitivity of the CRISPR/Cas system only reaches the pM level, and it is often necessary to use pre-signal amplification methods or engineered circuits to extend the detection limit [75,76]. Among them, the HCR-CRISPR strategy has been used to detect various targets. Cas12a can recognize dsDNA and initiate trans-cleavage activity under the guidance of crRNA but requires a pro-spacer adjacent motif (PAM) sequence adjacent to the target dsDNA, which may limit the broad application of its HCR. To address these issues, Liu et al. introduced the PAM sequence into the amplification product of HCR to construct a sensitive and rapid electrochemical assay for the detection of *S. Typhimurium* in milk [72]. In the presence of *S. Typhimurium*, Aptamer on IMB preferentially binds to the target. The naked Linker strand binds to the trigger strand of the HCR product that forms many ssDNA side chains, which activates the trans-cleavage activity of the Cas12a-crRNA complex without the PAM sequence, thereby cleaving the surface-modified electrical signal response of the Au electrode ssDNA reporter probes. When the target is absent, no trans cleavage activity of Cas12a is observed, thereby retaining the reporter probe on the surface of the Au electrode. The method allows for the selective and sensitive quantification of *S. Typhimurium* with a 20 CFU/mL LOD (Figure 7D).

To address the criticism of electrochemistry’s inherent instability and poor reproducibility [77], Wang et al. proposed a fluorescent-electrochemical dual-signal sensing strategy based on magnetic DNA-Walker and HCR coupled amplification for the detection of *Clostridium perfringens* (*C. perfringens*) [78]. The method can significantly improve accuracy and detect *C. perfringens* at a minimum concentration of 1 CFU/g in meat products (e.g., chicken, beef, duck, mutton, and pork) through a multi-modal signal output approach.

### 3.4. SERS Surface-Enhanced Raman Scattering

Surface-enhanced Raman spectroscopy (SERS) is also used in bioanalysis, including pathogen detection, because of its fingerprinting and interference-resistant, rapid, and high-resolution advantages [79]. So far, the HCR technique affects the SERS signal in two main ways: (1) an increase or decrease in the number of Raman molecules on the sensing substrate and (2) an aggregation or dispersion of Raman molecules on the sensing substrate.

Xu et al. proposed a direct SERE method for the detection of *S. aureus* in milk with a three-part system, Apt and cDNA modified gold magnetic receptive probes (Au-MNPs-dsDNA), 4-ATP modified gold core–silver shell signal response probes (Au@Ag/4-ATP) and an integrated PDMS conical sensing array [80]. The presence of the target allows the HCR reaction to take place and produce dsDNA containing sulfhydryl (-SH) groups, and Au@Ag/4-ATP is loaded in large quantities through a specific “Ag-S” bond. Magnetic aggregation capture with the PDMS array platform achieves stable SERS signal acquisition. Detection limits as low as 0.25 CFU/mL are achieved over a concentration range of 28 to 2.8 × 10^6^ CFU/mL for *S. aureus* (Figure 8A).

Li et al. instead proposed an indirect SERS strategy for the susceptible detection of *S. Typhimurium* in milk [81]. The HCR product was loaded with many Raman signal molecules DAPIs separated by IMB. Finally, the remaining DAPIs emitted a dense SERS signal due to electrostatic interactions with colloidal AgNP absorption and the formation of many hot spots. The results showed a detection limit of 6 CFU/mL within 3.5 h (Figure 8B).

In addition, the GQH DNAzyme has more than just peroxide-mimetic enzymatic catalytic activity; Xu et al. cleverly used its chemical reaction-catalytic ability (e.g., carbon–carbon bond formation, phosphorylation of organic hydroxyl groups, oxidation of organic molecules to generate chromatography, and oxidation of substrates such as thiols or anilines) to influence the extent of Raman molecular aggregation. The oxidation of L-cysteine to cystine (thiol oxidation to disulfide) is catalyzed by the HCR-generated GQH DNAzyme, which affects the L-cysteine-mediated aggregation of 4-NTP-modified AuNPs, increases the electromagnetic field, and amplifies the Raman signal. A wide linear range (5–5 × 10^5^ CFU/mL) and low detection limit (4 CFU/mL) for *S. Typhimurium* were achieved (Figure 8C). In addition, the method was designed with an improved split-blocking G-rich fraction of the hairpin probe, eliminating the possibility of self-assembly of the GQH DNAzyme in the absence of a target and significantly reducing the non-specific background signal [82].

### 3.5. HCR Combined with Portable Equipment for Point-of-Care Detection

It is worth noting that developing countries and economically backward regions usually have a high prevalence of foodborne diseases. Nevertheless, the HCR-based high-sensitivity strategies in the previous section often rely on a variety of sophisticated and sophisticated equipment (e.g., PCR instruments, fluorometers, electrochemical workstations, Raman spectrometers, etc.) as well as professional staff and laboratory conditions, etc. Therefore, developing low-cost, simple-to-operate, portable HCR-based high-sensitivity point-of-care (POC) solutions would benefit a wider region and population.

To increase sensitivity, Fang’s team developed the first HCR signal amplification-based lateral flow assay (HCR-LFA) for detecting *Salmonella* ssDNA in 2016 [83]. This method uses the traditional “sandwich-type” structure. In the presence of the bacteria’s DNA, a “FAM-CP/ssDNA/RP-dsDNA-biotin” bridge complex formed, which loads lots of streptavidin-modified AuNPs (AuNP-SA) when flowing through the Conjugate pad. Since the CP on the bridge complex is labeled FAM, it is finally captured by an anti-FAM monoclonal antibody on the T-line to form a red colloidal gold band. The detection limit for synthetic ssDNA is 1.76 pM, which is about two orders of magnitude lower than that of LFA without HCR amplification, and for detecting *Salmonella* is 3 × 10^3^ CFU/mL (Figure 9A).

Subsequently, Fang’s team investigated the extension of the application of other detection targets for foodborne pathogens based on the HCR-LFA strategy. Due to the tens of thousands of copies of 16S rRNA in individual bacterial cells, targeting them has a potential amplification effect. In 2020, Fang’s team proposed the HCR-LFA method for detecting *S. enteritidis* 16S rRNA in milk (Figure 9B). Probe hybridization was improved by dissociation of the secondary structure of 16S rRNA by multiple Helper probes. The method provides sensitive detection of *S. enteritidis* with a detection limit of 53.65 CFU/mL. At the same time, it solves the problem that immunoassays or DNA amplification-based methods cannot distinguish between live and dead bacteria [84]. However, to avoid cross-contamination, Wan’s team proposed an HCR-LFA strategy based on opening the 16S rRNA hairpin with the aid of an auxiliary probe and following an isothermal chain displacement reaction (ISD), thereby achieving displacement of the 16S rRNA [85] (Figure 9C).

However, for the detection of bacterial DNA/RNA, the extraction process is often complex and not easily preserved [86,87,88]. With the advent of nucleic acid aptamer technology, which has the potential to replace traditional protein antibody recognition, the conversion of non-nucleic acid targets (e.g., bacterial cells) to nucleic acid signals can be achieved, allowing for more sensitive and safe detection without contamination by nucleic acid amplification techniques such as HCR, which has excellent potential for development [89,90,91].

In 2021, Fang’s team used the HCR-LFA strategy to detect *V. parahaemolyticus* cells in shrimp based on the principle of competitive binding (Figure 10A). The pre-prepared HCR product containing a large number of biotin sites not only shortens the detection time but also allows for competitive binding, with Capture Probe on the T-line binding to Aptamer, leading to the dissociation of *V. parahaemolyticus*, avoiding the problem of large volumes of bacteria not being able to flow on the test strips. The method has a 2.6 × 10^3^ CFU/mL LOD, and the entire assay time is only 67 min [92]. Inspired by the idea of signal conversion, some POC detection strategies using HCR amplification to detect foodborne pathogens cleverly utilize commercial portable devices such as hCG test strips [93] and personal blood glucose meters [94] to display results, thus avoiding the preparation of complex LFA strips and simplifying operations.

The rapid development of microfluidic technology in recent years has enabled POC detection devices to be limited to those described above and laid the groundwork for constructing miniaturized and integrated rapid detection platforms for pathogenic bacteria. Li et al. present a microfluidic chip for naked-eye detection of *E. coli O157:H7* with integrated volumetric bar graphs (Figure 10B). The initiator was previously modified on the microfluidic plate, and the presence of a target triggered an HCR reaction loaded with a large number of PtNPs, which catalyzed the decomposition of H_2_O_2_ to produce O_2_ that drove the ink across the volumetric bar graph and performed a visual readout based on the distance to quantify the pathogen concentration. Rapid visual detection of *E. coli O157:H7* was achieved (75 min) with LODs of up to 250 and 400 CFU/mL in buffer and milk samples, respectively [95]. Chen et al. proposed a three-dimensional (3D) chip with a multi-channel structure for two types of pathogenic bacteria detection based on HCR [96]. The chip can ignore complex operations such as sample preparation and achieve simplified analysis by switching between multiple targets and multiple modes. In Mode 1, a long HCR amplification probe is used for “sandwich”-type bacterial detection to improve detection sensitivity; in Mode II, the inner surface of the 3D chip can serve as a substrate for rapid HCR assembly, shortening the detection time. The system can achieve whole-cell detection, and under Mode 1, the LOD for *S. aureus* is 4 CFU/mL; under Mode 2, the LOD for *Salmonella* is 8 CFU/mL (Figure 10C).

### 3.6. Others

Other sensitive types of HCR-based signal transduction platforms (e.g., SPR, chemiluminescence, electrochemiluminescence, etc.) for foodborne pathogen detection are gradually being proposed and still hold great promise for research.

Xia et al. developed an SPR approach for the rapid, sensitive detection of three pathogenic bacteria based on autocatalytic multi-component DNAzyme (MNA zyme) cyclical cleavage and HCR amplification [97]. When target DNA is present, the hairpin probe P1 is opened and binds to the target and H2 to form a cleaved MNA zyme. The specially designed substrate probe H0 is cleaved into two segments, of which cleaved segment 1 has a similar sequence to the target, thus continuing to cycle open H1 to form a new MNA zyme, improving amplification efficiency and cleaved segment 2, enriched by magnetic separation, acts as the SPR sensing interface connector and is captured by immobilized probes to form a long dsDNA nanowire product by HCR reaction of the exposed tails. Finally, the target DNA concentration is determined indirectly by detecting the resonance angular shift, representing the change in refractive index of the sensing membrane, with detection limits of 67 CFU/mL for *S. aureus*, 57 CFU/mL for *Klebsiella pneumoniae*, and 61 CFU/mL for *E. coli*, respectively (Figure 11A).

The Zhang group developed two luminescence protocols with miniaturized and integrated features for pathogenic bacteria detection based on cloth-based microfluidics [98,99]. The chemiluminescence strategy utilizes pre-generated HCR-GQH DNAzyme products to avoid the drawbacks of long reaction time and poor traditional enzyme stability. When the target ssDNA is present, the ssDNA serves as a bridge to link HCR-GQH DNAzyme products to the cloth-based microfluidic substrate and catalyzes the luminescent substrate luminal/H_2_O_2_ to produce a CL signal. This protocol achieved a detection limit of 50 CFU/mL in spiked milk samples (Figure 11B). In electrochemiluminescence, the strategy avoids the use of PCR. Instead, it uses a simple restriction endonuclease reaction to obtain *E. coli O157:H7* ssDNA, which is then amplified with multiple HCRs at multiple sites with the help of an auxiliary probe. Direct insertion of Ru(bpy)_3_^2+^ further enhances the ECL signal, achieving a low LOD of 38 CFU/mL and successfully detecting *E. coli O157:H7* in milk samples (Figure 11C).

## 4. Summary, Challenges, and Perspectives

In this review, the development of food safety detection technologies (especially for foodborne pathogens) is first briefly described. The basic principles, types, and technical characteristics of HCR are then described. Finally, the application of HCR amplification techniques for detecting foodborne pathogens over the last five years is discussed, with a focus on specific examples, summarized in Table 1. In general, the excellent amplification capabilities of HCR are demonstrated by the use of various signal transduction platforms [100] (e.g., fluorescent, colorimetric, electrochemical, etc.), which enable the sensitive detection of various markers (DNA, 16S rRNA, Cell) of pathogenic bacteria. Although HCR amplification technology has led to a significant increase in the sensitivity of the detection of foodborne pathogens, it has been accompanied by severe challenges.

The primary challenge is the rational design of HCR hairpins [31]. The length and binding strength of the toe sequence affects the reaction hybridization dynamics. To effectively initiate HCR, the toe point length is usually at least one-third of the stem length. Longer stem lengths and lower free energies facilitate the stability of the hairpin structure and avoid leakage, but also increase the difficulty of being opened. However, the success rate of the hairpin design has improved with non-specific software such as NUPACK and a four-point guide established by Yung’s group to guide the rational design of DNA hairpins concerning toe point, stem domain length, and GC content [101]. However, the increasing variety of advanced HCRs involving more pairs or complex hairpin designs is still blind to their design. There is still a pressing need to explore and summarize comprehensive HCR design guidelines for various types of HCR and software specifically designed for HCR sequence design [102]. Secondly, HCR usually designs the starting strand based on the target nucleic acid. Hairpin probes need to vary with different target molecules, and there need to be universal and efficient HCR signal amplification systems. Finally, the end of the HCR reaction can only be terminated by the depletion of the fuel strands; it is uncontrollable, and uncontrollable false positive amplification may occur. Coupled with complex matrix interference in food, controlled dynamic assembly of HCR remains elusive. Although some HCR target detection strategies using near-infrared light control and pH response have been reported in the field of bioimaging, there is still a long way to go for controllable, sensitive, and high-throughput controllable HCR techniques applicable to the detection of pathogenic bacteria in the actual food industry.

### Directions for Development

HCR is still more focused on bacterial DNA/RNA detection in foodborne pathogen detection due to its amplification properties based on nucleic acids. However, this often involves tedious operational steps such as cell lysis, target extraction, and preservation. With the continuous development of aptamer technology, various types of aptamers with particular recognition of foodborne pathogenic bacteria have emerged, making HCR more convenient, effective, and promising for the direct detection of whole bacterial cells, and further research should focus more on the detection of intact target bacteria to reduce complex operations and shorten detection time, and to extend the application to a broader range of pathogenic bacteria and actual samples.In food testing, the main focus is still on the rapid, accurate, and sensitive identification of foodborne pathogens. Sensitive “real-time monitoring” and “timely eradication” of pathogen loads during food production, processing, and distribution (e.g., pasteurization in milk) is essential to ensure food quality, shelf-life stability, economic cost savings, and protection of human life and health. The programmable and easily modifiable capabilities of HCR, combined with the evolving nanomaterials technology for sensitive identification, real-time monitoring, and timely eradication, are also areas of great potential for future exploration, requiring pioneers to continue to forge ahead.The simultaneous detection of multiple targets is critical in samples where more than one bacterium is often present. At present, HCR technology for pathogen detection is still dominated by a single strain. Although multi-target simultaneous detection methods have been proposed—for example, Lai’s group proposed a fluorescent HCR-ELISA method [103] based on the previous HCR-ELISA single-component colorimetric strategy [32] to achieve simultaneous detection of three pathogens—they are still relatively few and need to expand continually. In addition, although a single detection mode can achieve high sensitivity and specificity and often cannot avoid background interference and false positive problems, a multi-mode output approach with reasonable accuracy is a good solution and still has a broad research space.Finally, the amplification capacity of HCR is improving and even surpassing the amplification capacity of PCR technology. Due to its enzyme-free and mild conditions, it has a vast market and research potential in POC strategy and product development. It is important to note that the detection of pathogenic bacteria usually involves three steps: identification, signal amplification, and signal output. In contrast, HCR can act as an intermediate bridge, unifying the magnetic enrichment pre-processing technology with various signal transduction output platforms. Powered by microfluidic technology, it shows extraordinary potential for extracting and separating analytes from complex sample matrices for high-throughput one-plot or full-process detection. Sample-answer assays are bound to be a trend in the future and therefore require attention to research into miniaturized, integrated, and automated analytical systems. We expect more HCR-based POC strategies to emerge for foodborne pathogen detection for food safety.

## Figures and Tables

**Figure 1 foods-12-04067-f001:**
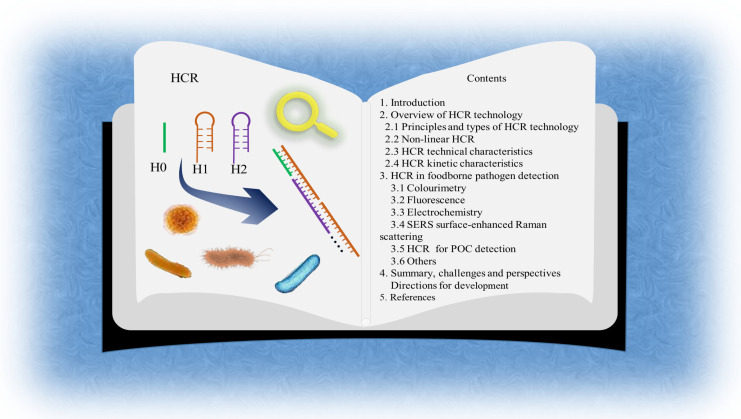
Basic principles of the HCR process and table of contents of the chapters.

**Figure 2 foods-12-04067-f002:**
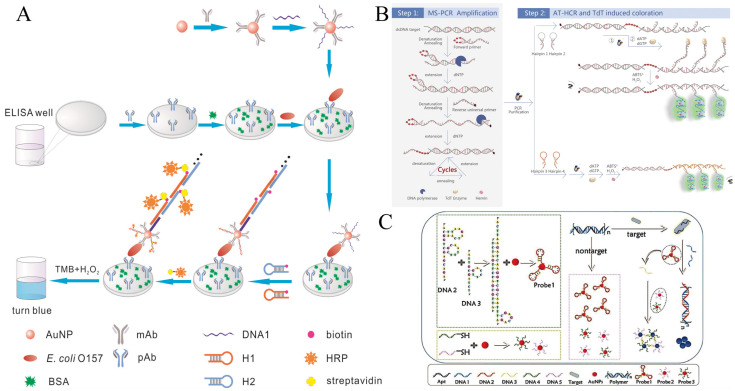
HCR-based colorimetric sensing platform for foodborne pathogen detection. (**A**) Sandwich type of ELISA using HCR combined with natural enzymes. Reprinted with permission from ref. [32]. Copyright (2016) Elsevier. (**B**) Dual amplification using PCR-HCR combined with GQH DNAzyme. Reprinted with permission from ref. [33]. Copyright (2018) Elsevier. (**C**) AuNP-based colorimetric sensing combined with HCR strategies works mainly through two mechanisms. Reprinted with permission from ref. [34]. Copyright (2022) American Chemical Society.

**Figure 3 foods-12-04067-f003:**
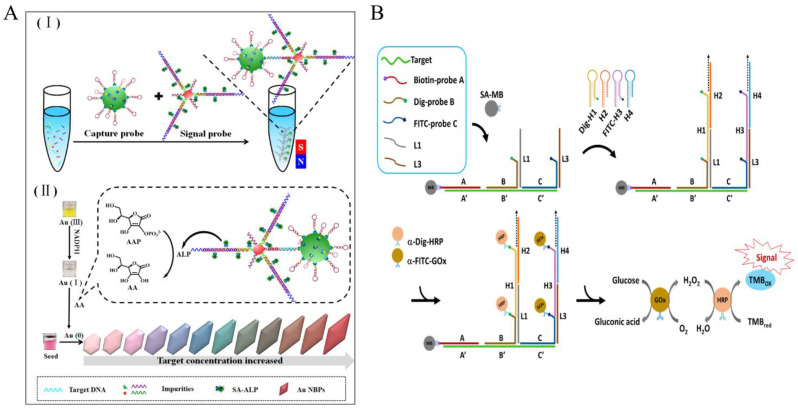
HCR-based colorimetric sensing platform for foodborne pathogen detection. (**A**) Multi-color colorimetric strategy using multi-HCR combined with gold nanobipyramids. (I) the hybridization process of capture probe, target DNA and signal probe; (II) the signal producing process by using NADPH-assisted ALP-mediated in situ growth of AuNBPs. Reprinted with permission from ref. [38]. Copyright (2021) Elsevier. (**B**) DNA logic-gate circuits using HCR combined with enzymatic cascade amplification. Reprinted with permission from ref. [39]. Copyright (2022) Elsevier.

**Figure 4 foods-12-04067-f004:**
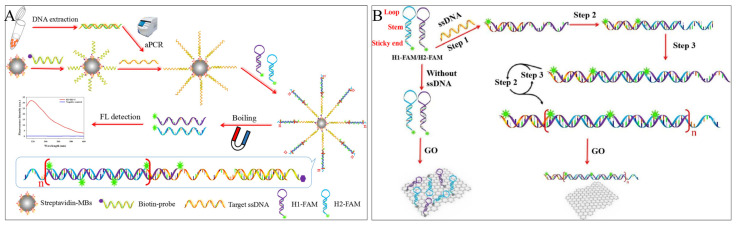
HCR-based fluorescence sensing platform using the carrier for foodborne pathogen detection. (**A**) A direct method used with magnetic beads. Reprinted with permission from ref. [41]. Copyright (2018) Elsevier. (**B**) A “HCR-GO” platform. Reprinted with permission from ref. [44]. Copyright (2021) Elsevier.

**Figure 5 foods-12-04067-f005:**
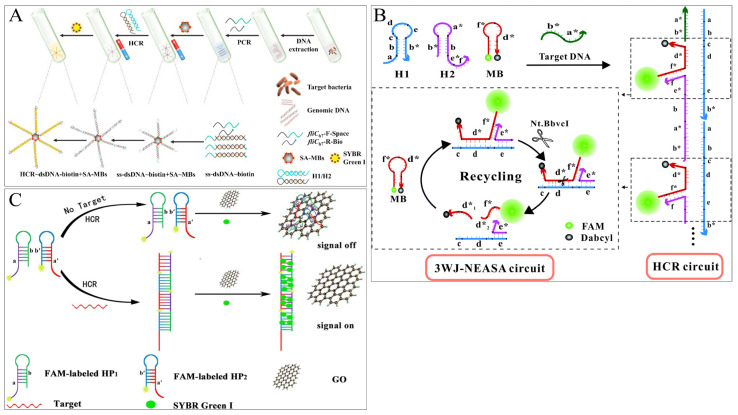
HCR-based fluorescence sensing platform using various auxiliary collaborative amplification strategies for foodborne pathogen detection. (**A**) Multi-technology amplification. Reprinted with permission from ref. [46]. Copyright (2020) Elsevier. (**B**) Enzyme-assisted HCR. Reprinted with permission from ref. [47]. Copyright (2021) Royal Society of Chemistry. (**C**) Synergistic effect. Reprinted with permission from ref. [48]. Copyright (2019) Elsevier.

**Figure 6 foods-12-04067-f006:**
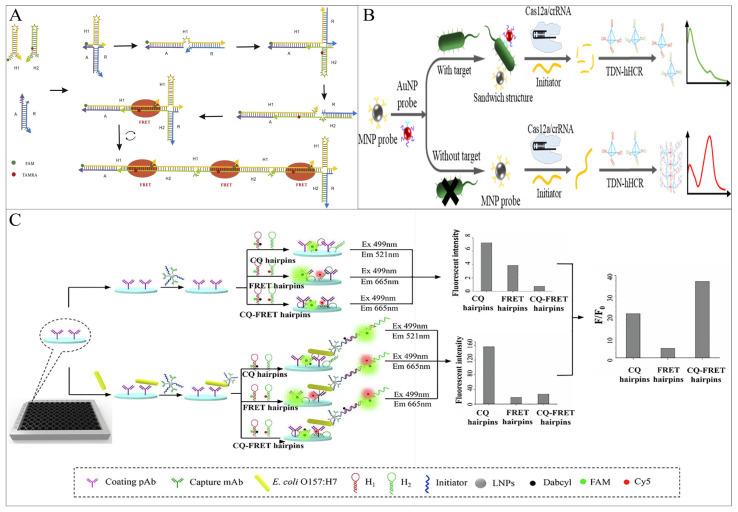
HCR-based fluorescence sensing platform using energy transfer strategies for foodborne pathogen detection. (**A**) A simple FRET-based method using a four-way migration HCR. Reprinted with permission from ref. [51]. Copyright (2019) Elsevier. (**B**) A complex FRET-based method using tetrahedral DNA nanostructure-mediated HCR combined with CRISPR-Cas12a. Reprinted with permission from ref. [53]. Copyright (2022) American Chemical Society. (**C**) An ELISA method combining FRET and CQ. Reprinted with permission from ref. [54]. Copyright (2020) Elsevier.

**Figure 7 foods-12-04067-f007:**
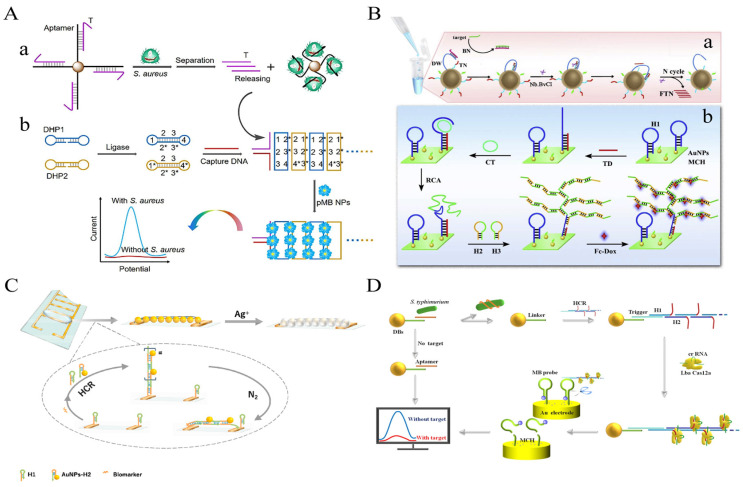
HCR-based electrochemical sensing platform for foodborne pathogen detection. (**A**) Dumbbell HCR combined with poly methylene blue nanoparticles. Reprinted with permission from ref. [69]. Copyright (2022) Elsevier. (**B**) Dendritic HCR combined with Fc-Dox. (a) the schematic diagram of the 3D DNA walker-based amplification reactions triggered by the target gene to produce many fragment of TN (FTN); (b) the schematic diagram of HCR and RCA-based amplification reactions on the electrode surface to generate to long double-stranded DNA sequences to immobilize many electrochemical indicators related with the concentration of target gene. (AuNPs, gold nanoparticles; BN, Blocking DNA; CT, Circular template; DW, DNA walker; FTN, fragment of TN; H1, hairpin DNA 1; H2, hairpin DNA 2; H3, hairpin DNA 3; HCR, hybridization chain reaction; MCH, 6-mercapto ethanol; RCA, rolling circle amplification; TN, transfer oligonucleotide) Reprinted with permission from ref. [70]. Copyright (2019) Elsevier. (**C**) A facile synthesis of silver wire on the electrode combined with AuNPs. Reprinted with permission from ref. [71]. Copyright (2020) Elsevier. (**D**) The reporter probe on the surface of the electrode was cleaved by HCR-CRISPR/Cas12a system. Reprinted with permission from ref. [72]. Copyright (2021) Springer Nature.

**Figure 8 foods-12-04067-f008:**
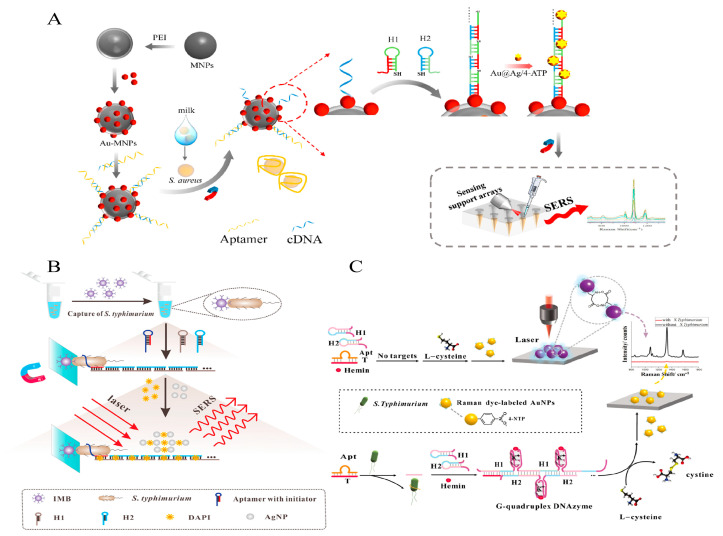
HCR-based SERS sensing platform for foodborne pathogen detection. (**A**) A direct method based on the combination of HCR and SERS. Reprinted with permission from ref. [80]. Copyright (2022) Elsevier. (**B**) An indirect method based on the combination of HCR and SERS. Reprinted with permission from ref. [81]. Copyright (2020) Elsevier. (**C**) A method for modulating Raman molecular states by HCR self-assembly. Reprinted with permission from ref. [82]. Copyright (2022) Springer Nature.

**Figure 9 foods-12-04067-f009:**
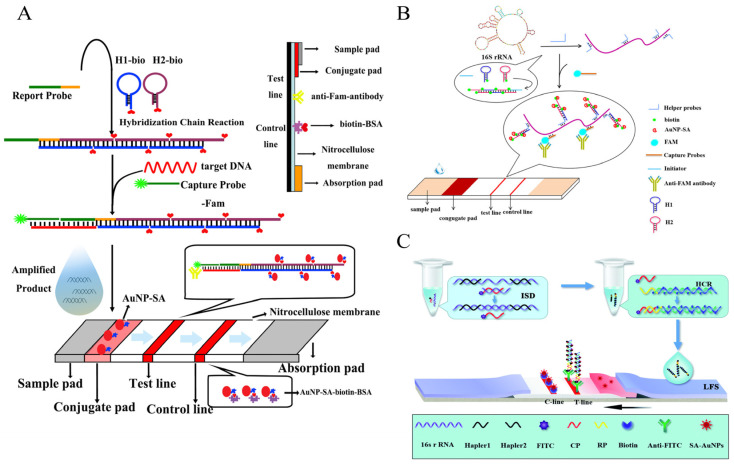
HCR-based POC strategy using lateral flow analysis for DNA/RNA detection of foodborne pathogens. (**A**) A ssDNA detection method using typical HCR. Reprinted with permission from ref. [83]. Copyright (2016) Elsevier. (**B**) A 16S rRNA detection method using a helper probe to expose secondary structure. Reprinted with permission from ref. [84]. Copyright (2020) John Wiley and Sons. (**C**) A 16S rRNA detection method combining HCR and ISD amplification. Reprinted with permission from ref. [85]. Copyright (2021) RSC Publishing.

**Figure 10 foods-12-04067-f010:**
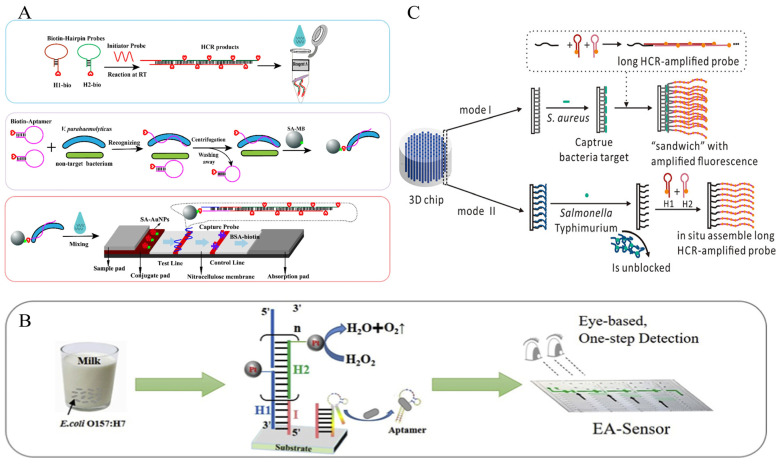
HCR-based POC strategy for whole-cell detection of foodborne pathogens. (**A**) A lateral flow analysis combining HCR and MBs. Reprinted with permission from ref. [92]. Copyright (2021) Springer Nature. (**B**) HCR combined with a microfluidic chip with a volume bar chart. Reprinted with permission from ref. [95]. Copyright (2020) Elsevier. (**C**) HCR combined with a multi-mode 3D chip. Reprinted with permission from ref. [96]. Copyright (2018) American Chemical Society.

**Figure 11 foods-12-04067-f011:**
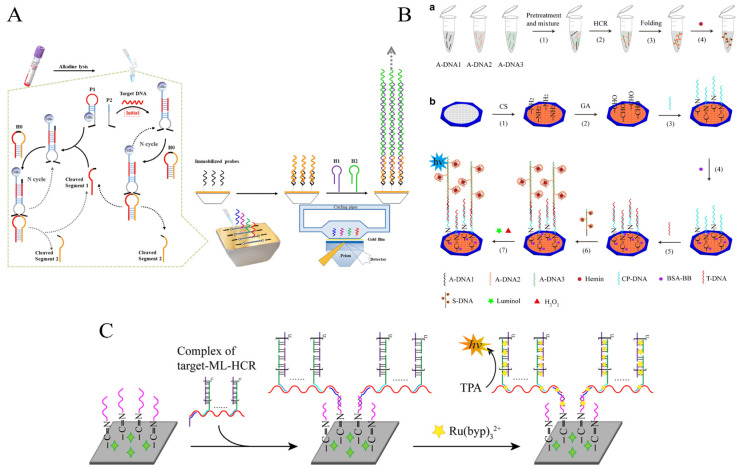
HCR-based other sensing platforms for foodborne pathogen detection. (**A**) The SPR strategy combined with autocatalytic multi-component DNAzyme. Reprinted with permission from ref. [97]. Copyright (2020) Elsevier. (**B**) Cloth-based chemiluminescence strategy. Reprinted with permission from ref. [98]. Copyright (2020) Springer Nature. (**C**) Cloth-based electrochemiluminescence strategy. Reprinted with permission from ref. [99]. Copyright (2022) Elsevier.

**Table 1 foods-12-04067-t001:** HCR-based biosensors for foodborne pathogen detection.

Signal Transduction	Detected Pathogens and Target Type	Strategy	Linear Range	LOD	Detection Time	Food Samples	Ref.
Colorimetry	*E. coli O157:H7*; cell	ELISA; HRP enzyme	5 × 10^2^–1 × 10^7^ CFU/mL	1.08 × 10^2^ CFU/mL in pure culture and 2.6 × 10^2^ CFU/mL in milk	~7.5 h	Milk	[32]
	*Salmonella* spp. and *S. aureus*; DNA	G-quadruplex DNAzyme	1–1 × 10^5^ CFU/mL	10 CFU/mL	130 min	Skim milk	[33]
	*Escherichia coli*; cell	Two types of mechanisms based on AuNP	1 × 10^2^–1 × 10^4^ CFU/mL	10 CFU/mL	~80 min	Tap water and milk	[34]
	*S. aureus*; *mecA* gene	Multi-color; AuNBPs	10–100 pM	2.71 pM	1 h	Milk	[38]
	*L. monocytogenes*; genomic DNA	GOx/HRP enzyme cascade; Boolean logic function	0.1–10 nM (colorimetry); 0−50 nM (electrochemistry)	1.12 nM (colorimetry); 0.04 nM (electrochemistry)	2 h	Vegetable, meat, and milk extracts	[39]
Fluorescence	*S. enteritidis*; *invA* gene	MBs; hairpin labeled FAM	7.4 × 10^1^–7.4 × 10^8^ CFU/mL	74 CFU/mL	~2.5 h	Lettuce	[41]
	*Mycobacterium tuberculosis*; *IS6110* gene	MBs; hairpin labeled TAMRA; turn-off	0.01–100 nM	10 pM	~2 h	N/A	[42]
	*emetic Bacillus cereus*; genomic DNA	MBs; hairpin labeled FAM; flow cytometry	7.6–7.6 × 10^6^ CFU/mL	7.6 CFU/mL in buffer and 9.2 × 10^2^ CFU/mL in milk	~4.5 h	Milk	[43]
	*S. enteritidis*; *invA* gene	“GO fluorescence” platform; hairpin labeled FAM; turn-on	4.2 × 10^1^–4.2 × 10^7^ CFU/mL	4.2 × 10^1^ CFU/mL in buffer and 4.2 × 10^2^ CFU/mL in milk	3.5 h	Milk	[44]
	*E. coli O157:H7*; *fliCh7* gene	“PCR-HCR” dual-signal amplification; SYBR Green I embed dsDNA	7.2 × 10^1^–7.2 × 10^6^ CFU/mL	7.2 × 10^1^ CFU/mL in buffer and 7.2 × 10^2^ CFU/mL in milk	~3.5 h	Milk	[46]
	*S. aureus*; DNA	“HCR-3WJ-NEASA” circuit dual-signal amplification; CQ	10 pM–10 nM	6.7 pM	0.5 h	Milk	[47]
	*S. aureus*; 16S rRNA	“FAM-SYBR Green I” cooperative effect; “GO fluorescence” platform	50 pM–100 nM	50 pM	~1.5 h	Milk	[48]
	MRSA; *agrA* gene transcription (its mRNA)	“FAM-SYBR Green I” cooperative effect; “GO fluorescence” platform; strand-displacement polymerization reaction (SDPR)	10 fM–100 pM	10 fM	~1 h	N/A	[49]
	MRSA; *agrC* gene transcription (its mRNA)	“FAM-SYBR Green I” cooperative effect; “GO fluorescence” platform; Nb.BbvcI-assisted target recycling amplification (NATR)	10 fM–100 pM	7.5 fM	50 min	N/A	[50]
	*Vibrio Parahaemolyticus*; *gyrB* gene	FRET; four-way branch migration HCR circuits	0.2–60 nM	0.067 nM	90 min	N/A	[51]
	*S. Typhimurium*; cell	FRET; ratio fluorescence; CRISPR-Cas12a system and TDN-hHCR	10–10^8^ CFU/mL	8 CFU/mL	~2.5 h	Milk and egg white	[53]
	*E. coli O157:H7*; cell	FRET; ratio fluorescence; CQ	4.9 × 10^1^–4.9 × 10^6^ CFU/mL	3.5 × 10^1^ CFU/mL	~3 h	Milk	[54]
	*S. aureus*; cell	IFE; UCNPs; g-C_3_N_4_ NSs	10–10^6^ CFU/mL	1 CFU/mL	3 h	Tap water and milk	[57]
Electrochemistry	*S. aureus*; cell	pMB NPs; DHCR	10–10^8^ CFU/mL	1 CFU/mL	~3 h	Milk and pear juice	[69]
	*E. coli O157:H7*; cell	Fc-Dox; multiple amplification through the 3D DNA walker, RCA, and HCR	10–10^4^ CFU/mL	7 CFU/mL	~2 h	N/A	[70]
	*S. aureus*; 16S rRNA	Silver wire across electrodes; hairpin labeled AuNPs	50–10^7^ CFU/mL	50 CFU/mL	100 min	Milk	[71]
	*S. Typhimurium*;cell	CRISPR-Cas12a; electrode surface modification of electrical response reporting probes	10^4^–10^8^ CFU/mL	20 CFU/mL	~2.5 h	Milk	[72]
	*Clostridium Perfringens*; DNA	DNA-Walker; methylene blue; dual-mode output	1–10^8^ CFU/g	1 CFU/g	N/A	Chicken, beef, duck, mutton, and pork	[78]
Surface-enhanced Raman scattering	*S. aureus*; cell	Au@Ag/4-ATP; PDMS-based SERS platform	28–2.8 × 10^6^ CFU/mL	0.25 CFU/mL	2.5 h	Milk	[80]
	*S. Typhimurium*; cell	AgNP/DAPI; competitive indirect strategies	10–10^5^ CFU/mL	6 CFU/mL	3.5 h	Milk	[81]
	*S. Typhimurium*; cell	4-NTP/AuNPs; G-quadruplex DNAzyme	5–10^5^ CFU/mL	4 CFU/mL	105 min	Milk and tap water	[82]
POC test	*S. enteritidis*; DNA	Lateral flow; Sandwich structure	2.5 pM–500 nM	1.76 pM	~25 min	N/A	[83]
	*S. enteritidis*; 16S rRNA	Lateral flow; “initiators-on-a-string”complex	10^2^–10^4^ CFU/mL	53.65 CFU/mL	~40 min	Milk	[84]
	*E. coli O157:H7*; 16S rRNA	Lateral flow; ISD	10^2^–10^5^ CFU/mL	10^2^ CFU/mL	~1.5 h	Milk	[85]
	*Vibrio parahaemolyticus*; cell	Lateral flow; MBs; sandwich structure	10^3^–10^8^ CFU/mL	2.6 × 10^3^ CFU/mL	67 min	Shrimp	[92]
	*E. coli O157:H7*; cell	Microfluidic chip; PtNPs	5 × 10^2^–5 × 10^7^ CFU/mL	250 CFU/mL in buffer and 400 CFU/mL in milk	75 min	Milk	[95]
	*S. aureus*; *S. enteritidis*	Microfluidic chip; multi-mode analysis	3.6 × 10^1^–3.6 × 10^6^ CFU/mL	4 and 8 CFU/mL, respectively	15 min	Nonfat milk powder and raw ground pork	[96]
	*E. coli O157:H7*; cell	Pregnancy test strips and MOF; sandwich structure	10^3^–10^7^ CFU/mL	530 CFU/mL	86 min	Milk	[93]
	*S. aureus*; cell	Personal glucose meter; the invertase catalyzes sucrose into glucose	3–3 × 10^3^ CFU/mL	2 CFU/mL	4.5 h	Peach juice, milk, and water samples	[94]
Others							
Surface plasmon resonance (SPR)	*S. aureus*, *Klebsiella pneumoniae* and *Escherichia coli*; DNA	MNAzyme	1 × 10^2^–1 × 10^6^ CFU/mL	67 CFU/mL of *S. aureus*, 57 CFU/mL of *K. pneumonia* and 61 CFU/mL of *E. coli*	4 h	N/A	[97]
Chemiluminescence (CL)	*L. monocytogenes*; *hlyA* gene	Cloth-based microfluidics; G-quadruplex DNAzyme	2 × 10^−3^–2 × 10^6^ pM	1.1 fM	80 min	Milk	[98]
Electrochemiluminescence (ECL)	*E. coli O157:H7*; genomic DNA	Cloth-based microfluidics; Ru(bpy)_3_^2+^	10^2^–10^7^ CFU/mL	38 CFU/mL	~100 min	Milk	[99]

Note: N/A means not available.

## Data Availability

The data used to support the findings of this study can be made available by the corresponding author upon request.

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
