# Peer review of "The Application of Hybridization Chain Reaction in the Detection of Foodborne Pathogens"

_foods, 2023, doi:10.3390/foods12224067_

Round 1

Reviewer 1 Report

1.      Introduction lack some important information about foodborne bacterial pathogens. Please improve it.

2.      Introduction section - line 24 – (Bacillus Ceres) not correctly spelled.

3.      Introduction section - Line 28 – Polymerase chain reaction (PCR) –  already abbreviated above.

4.      Overview of HCR technology – First proposed by Dirks. What is proposed? Add HCR.

5.      Heading 3 line 1. You started the sentence with “In general” - which is a conclusive term. Please revise this section. It would be good if you add some important documented findings about detection of foodborne pathogens related to HCR.

6.      Subheading 3.1. Please revise the first line – I recommend you first to describe about what colorimetry means.

7.      Page 5. Second paragraph you can shorten the last statement (10 CFU/mL for both salmonella and S. aureus). Also, which salmonella serovar? Which food matrices?

8.      In most cases you have mentioned the detection limit (concentration) for different types of pathogen detection you have revised however, detection time or assay time is missed including your table.  If possible, try to include it.

9.      Subheading 3.3. the first paragraph lack reference. Do you think the term “Electrochemistry” refers to sensor. Please define properly or use appropriate terminologies.

10.  Have you described the term CRISPR?

11.  Page 12 – S. typhimurium. Check how to write serotypes.

12.  Page 12 – avoid using abbreviations in heading and subheadings (Change POC – point-of-care).

13.  Page 14 – From  however up to development – lack reference.

14.  You have revised some of the detection approaches without indicating the food matrices. If you have encountered these data, please incorporate it.

15.  Page 15 – subheading 3.6. You started the sentence with “in addition” which not good. It seems some missing information. Revise it. You could start it without “in addition”

It would be better to incorporate some of the good values of HCR technologies in comparison to the other foodborne pathogens detection methods.

Thank you.

Best wishes.

Author Response

Dear editors and reviewers,

thank you for your valuable comments.  Please see the attachment for specific response. 

Best wishes 

Reviewer 2 Report

General comments: The authors have a deep understanding of HCR, but do not seem to understand food safety from an international perspective. In many parts of the world, the current food safety monitoring system is doing a good job and people have confidence in the safety of the food supply.  There are currently a number of antimicrobial hurdles which have been established to maintain food safety.   The tone of this manuscript suggests that current food safety practices are largely ineffective and cases of foodborne disease are alarming and out of control.  The current system for food safety can definitely be improved, but the paper needs to be modified to be less critical as the HCR techniques described are also not perfect.

Individual comments:

Abstract - Why is the food trade alarming?  Need to re-phrase and provide some examples of recent foodborne outbreaks of "glaring" prominence

Introduction - 'Highly prevalent' - what does that mean?  Please define by cases per 100,000 population or cite the impact of some recent outbreaks

ICP-MS   need to define

P2L1 which remain present

P2L3 this section needs to be re-written

P2 paragraph 2  Not all Elisas have poor sensitivity.  Some including those for foodborne pathogens work well.   You propose using ELISA in conjunction with HCR later, so probably good idea not to completely discount utility of ELISA here.  RPA is recombinanse polymerase amplification

Section 2.2 define dsDNA

Last line on P6 - what is the limit of detection.  Not so useful to know what it is not.

P7 - need to define ssDAN

Table 1 would suggest 'Food Samples' instead of 'Real Samples'

Throughout the manuscript - please italicize bacterial genus and species names appropriately.

P19 PH = pH???, NIR = near infrared?
P20 POC = ???

This manuscript would benefit from editing for English usage. Some sections read well, while others are difficult if not impossible to understand

Author Response

(The authors gave the same response as above.)
